# Analysis of Predictors and Risk Factors of Postpolypectomy Syndrome

**DOI:** 10.3390/diagnostics14020127

**Published:** 2024-01-05

**Authors:** Stefano Fusco, Michelle E. Bauer, Ulrike Schempf, Dietmar Stüker, Gunnar Blumenstock, Nisar P. Malek, Christoph R. Werner, Dörte Wichmann

**Affiliations:** 1Department of Internal Medicine I, Section of Gastroenterology, Gastrointestinal Oncology, Hepatology, Infectiology and Geriatrics, University Hospital of Tübingen, 72076 Tübingen, Germanyulrike.schempf@med.uni-tuebingen.de (U.S.); nisar.malek@med.uni-tuebingen.de (N.P.M.); christoph.werner@med.uni-tuebingen.de (C.R.W.); doerte.wichmann@med.uni-tuebingen.de (D.W.); 2Department of Clinical Epidemiology, Eberhard-Karls-University, 72076 Tübingen, Germany

**Keywords:** postpolypectomy syndrome, PPS, colonoscopy, polyp, adenoma, abdominal pain

## Abstract

Background and aims: Postpolypectomy syndrome (PPS) is a relevant adverse event that can appear after polypectomy. Several publications mention postpolypectomy syndrome using different criteria to define it. The aim of this study is to detect potential risk factors and predictors for developing PPS and to define the main criteria of PPS. Methods: In this retrospective monocentric study, 475 out of 966 patients who underwent colonoscopy with polypectomy from October 2015 to June 2020 were included. The main criterion of PPS is defined as the development of postinterventional abdominal pain lasting more than six hours. Results: A total of 9.7% of the patients developed PPS, which was defined as local abdominal pain around the polypectomy area after six hours. A total of 8.6% of the study population had abdominal pain within six hours postintervention. A total of 3.7% had an isolated triad of fever, leukocytosis, and increased CRP in the absence of abdominal pain. Increased CRP combined with an elevated temperature over 37.5 °C seems to be a positive predictor for developing PPS. Four independent risk factors could be detected: serrated polyp morphology, polypoid configurated adenomas, polyp localization in the cecum, and the absence of intraepithelial neoplasia. Conclusions: Four independent risk factors for developing PPS were detected. The combination of increased CRP levels with elevated temperature seems to be a predictor for this pathology. As expected, the increasing use of cold snare polypectomies will reduce the incidence of this syndrome. Key summary: Our monocentric study on 966 patients detected four independent risk factors for developing PPS: pedunculated polyp, resected polyps in the cecum, absence of IEN, and serrated polyp morphology. The combination of increased CRP levels with elevated temperature seems to be a predictor for this pathology.

## 1. Introduction

The definition of postpolypectomy syndrome (PPS) is very heterogeneous in the literature [1,2,3]. The main criteria to define PPS vary from one single criterion (like abdominal pain) to multiple criteria (abdominal pain combined with at least one inflammation marker) that must be met to fulfill the diagnosis of PPS. There are many different nomenclatures describing PPS like “coagulation syndrome”, “postpolypectomy coagulation syndrome”, or “post ESD coagulation syndrome”, to name just a few of them. At present, we are not aware of any national or international guidelines for the diagnosis and/or management of PPS.

PPS is a poorly understood condition after polypectomy and little is known about the risk factors [4,5]. The difficulty in recognizing PPS is due to the similar symptoms of a postinterventional (micro)perforation, which include abdominal pain and local peritonism [6].

The aim of this monocentric, retrospective study is to evaluate certain independent risk factors that are associated with the development of PPS and some clinical predictors that increase the awareness of possible PPS [5,7]. In clinical practice, it is quite common to search for an infectious disease using blood cultures, ultrasound, or CT scan when patients develop fever or show increased inflammation markers like CRP or leukocytosis. As well as colon perforation and colon bleeding, PPS is one of the most important adverse events after endoscopic polypectomy [8].

PPS was first mentioned in a publication by Rogers et al. in 1975, which described PPS as transmural burn [9]. Waye et al. described a postpolypectomy coagulation syndrome in 1981 [10]. To our knowledge, the number of potential risk factors and predictors of PPS found in our retrospective trial has not been published yet.

## 2. Methods

### 2.1. Study Setting and Patient Selection

The local Institutional Review Board approved this study (IRB number: 833/2019BO2). All patients treated in the period between October 2015 and June 2020 in our central endoscopy unit who underwent endoscopic polypectomy were considered for inclusion in this retrospective study, given the following criteria were fulfilled: 18 years of age or older, inpatient care, documentation of vital parameters and pain score, documentation of blood values (e.g., hemoglobin, CRP, leukocytes), and colonoscopy with at least one resected polyp in the colon or rectum. Exclusion criteria were the following: co-infection, co-morbidities like IBD, hereditary polyposis syndromes, chronic pain, bleeding after polypectomy, outpatient care, irritable bowel syndrome, history of analgesic drugs (WHO class III), and other single-case decisions.

A total of 966 patients underwent colonoscopy with polypectomy in the screened period, as shown in Figure 1. They were detected via operating procedure codes (OPCs) in our hospital database (SAP). After removing duplicate OPCs, 177 patients were excluded from the study. A total of 314 patients fulfilled an exclusion criterion, so 475 patients were included in our trial. We examined different basic characteristics like age, sex, weight, BMI, nicotine abuse, pre-existing chronic diseases, and medication with analgesics or co-analgesics.

Due to the very heterogeneous definitions of PPS, in this analysis, PPS is defined as a postinterventional, local abdominal pain in the anatomical area of polypectomy that persists for at least six hours after the end of the colonoscopy [2,3,11].

### 2.2. Statistical Analysis

Statistical analysis was performed using the Statistical Software Package SPSS version 29.0 for Windows (IBM Corp, Armonk, NY, USA). For baseline descriptive statistics, means with standard deviation were used for variables with a parametric distribution, and medians with 25th and 75th percentiles were used for those with a nonparametric distribution. Outcome variables were reported as medians with adjusted 95% confidence intervals (CIs). The Mann–Whitney U test, the χ^2^-test, and Fisher’s exact test were used to compare groups where appropriate. The significance level was defined as *p* < 0.05. The primary endpoints of this monocentric, retrospective study were the analysis of the clinical course of PPS, as well as potential risk factors and predictors of PPS.

## 3. Results

A total of 475 patients who fulfilled the inclusion criteria were included, with a total number of 1289 endoscopic resected polyps. Forty-six patients (9.6%) developed PPS, defined as a postinterventional, local abdominal pain in the anatomical area of polypectomy, which persisted for at least six hours after the colonoscopy. A total of 133 polyps were resected in the PPS group. Baseline characteristics including demographic and clinical features, summarized in Table 1, did not significantly differ between the two cohorts. There was a dominance of the male sex in both cohorts. None of the pre-existing diseases seemed to be a risk factor for the development of PPS. Neither nicotine abuse nor BMI could predict PPS in our study cohorts. The rate of antibiotic administration was significantly higher in the PPS cohort; nonetheless, the antibiotic administration rate of approximately 23.9% in the PPS group was very low. The detection of bacteria, collected in blood culture samples, did not play a significant role, as almost all blood cultures were negative in the control group and none were positive in the PPS cohort.

In cases of suspected acute abdomen with increasing abdominal pain and a beginning of peritonism symptoms, a computed tomography scan was performed to detect or deny periluminal air, which corresponds to a micro-perforation. In both groups, just one patient had periluminal air after polypectomy. Adverse events were only recorded in the control group due to the exclusion criteria and are illustrated in Table 1.

Less than a quarter (23.9%) of the PPS cohort was administered an antibiotic treatment, while the control group received an antibiotic treatment in almost 10% of the cases (Table 1). None of the PPS cohort had a positive blood culture. Only in two blood cultures of the control group could the presence of bacteria be proven. The number of adverse events (AEs) was recorded only in the control group because the presence of the listed adverse events below was an exclusion criterion for the diagnosis of PPS. The most common AE was periinterventional bleeding, which happened in 12.4% and could be stopped during the endoscopic procedure. Postinterventional bleedings led to a second endoscopy and could be treated endoscopically as well. Perforations were as seldom as postpolypectomy fever (PPF), which appeared in 1.9 and 1.7%, respectively.

The endoscopic characteristics are summarized in Table 1. We mentioned the cohort sample size as well as the total polyp number in Table 2a,b because some features can be categorized for each single polyp. Apart from the higher rate of polyps located in the cecum in the PPS group, no further significant differences between both cohorts could be observed. The percentage of the result of resection and the method of intervention was nearly equal in both groups. The number of clip applications and the rate of relevant diverticulosis in the PPS group were similar to the control group. Vomiting, nausea, and gag reflex were subsumed as particularities in Table 2a.

Between the morphology of the polyps in both cohorts (tubular, tubulo-villous, villous), there are no significant differences, although the control cohort shows a dominance of the tubular morphology in almost half of the cases and a higher rate of the tubulo-villous polyps compared to the PPS cohort (16% vs. 10%). The PPS cohort has almost a three times higher rate of serrated polyp morphology than the control cohort (11% vs. 4%).

The polyp configuration was compared in both cohorts using the Paris classification (PC) as shown in Figure 2 and Figure 3. In the control cohort, the polypoid configuration, corresponding to PC 0-I, is predominant in the resected polyps with approximately 28.8%, followed by diminutive polyps with a relative rate of 19.3% and the non-polypoid configuration, according to PC 0-II, with a value of 16.5%. Within the polypoid polyps, sessile polyps (PC 0-Is) appear more often than pedunculated in the control cohort. The PPS cohort is predominated by pedunculated polyps (PC 0-Ip) with a share of 15.0% of the resected polyps. The same cohort has a similar proportion of polypoid polyps (PC 0-I) and non-polypoid polyps (PC 0-II) with 24.8% and 25.6%. While pedunculated polyps have the highest proportion of polyps with PC 0-I in the PPS cohort with 60.4%, sessile polyps (56.6%) are more often resected in the control cohort, compared to all polypoid polyps (PC 0-I) of the same cohort.

All diminutive polyps, defined as small flat or sessile polyps with a diameter of not more than 2 mm, were not categorized by the Paris classification, although they would normally correspond to PC 0-Is or 0-IIa. Instead of using the PC, we counted them separately, because we wanted to know whether diminutive polyps play a significant role in the development of PPS. But both cohorts demonstrated similar relative rates of diminutive polyps with about 19% each. If no specific PC was chosen for the polypoid or non-polypoid polyps, we classified those polyps as 0-I* or 0-II*.

Most of the results of the histological grading of the resected polyps are similar between both cohorts, e.g., the combined appearance of LGIEN and HGIEN in the same polyp, the HGIEN or the malignancy grading, and the missing values (Figure 4).

However, there is a relevant difference in the histological grading of resected polyps between the control cohort and the PPS subgroup: a large difference can be seen in the control cohort regarding the presence of an LGIEN (63.8%) and no IEN (31.3%). In the PPS group, no relevant difference could be detected between no IEN (44.4%) and LGIEN (48.9%). A significant difference could be noticed between both cohorts with resected polyps without an IEN (29.8 vs. 44.4%).

Comparing the localization of the resected polyps in the left side of the colon and the right side of the colon between both cohorts (Figure 5 and Figure 6), it is evident that left-sided colon polyps are more common in the control group (41.9% vs. 30.8%), while the PPS group is dominated by right-sided polyps (60.2% vs. 54.2%). From the six colorectal segments, the percentage of resected polyps in the colon ascendens, the transversum, and the sigmoid colon are nearly equal in both cohorts. The percentage of the rectal- and descendant-located resected polyps is twofold more frequent in the control cohort. The most significant difference between both cohorts turned out to be the location of the resected polyps in the cecum, which was around 25.6% in the PPS cohort and just 16.8% in the whole group. The presence of a polyp in the cecum is a risk factor for developing PPS.

Four potential risk factors for developing PPS could be detected using the univariate test. The localization of polyps in the cecum, a lateral-spread serrated morphology of polyps, a pedunculated configuration of polyps, and the absence of an intraepithelial neoplasia in the resected polyps showed a significant difference between both study groups (Table 3). On the other hand, pre-existing diseases, the duration of the procedure, and the resection in piecemeal technique did not differ between both cohorts significantly.

Multivariate linear regression (Table 4) demonstrates the independence of the four risk factors compared between the control cohort and the PPS group.

A total of 1289 polyps were resected. The polyp size of the whole study population ranged from <5 mm to more than 40 mm. Most of them, approximately half of all the polyps, had a size of 10 mm or less. A size greater than 30 mm was seldom found, with a rate of approximately 3%. The PPS cohort (133 resected polyps) consists of a similar range regarding the polyp size. Almost half of these resected polyps have a size of 10 mm or less. No significant difference in polyp size was observed between the two groups (Figure 7).

The number of resected polyps was quite equal in both groups; the higher the number of resected polyps was, the lower the relative rate was. Also, the duration of the endoscopy procedure during the polypectomy of the colorectum was not significantly different between both cohorts.

Table 5 shows the potential PPS predictors. Only an elevated CRP level is an independent predictor of PPS (calculated with a multiple linear regression), whereas the combination of elevated CRP levels and increased body temperature can better predict the development of PPS.

The inflammation markers mentioned in Table 6 are correlated with the duration of abdominal pain. The aim was to check whether there is a significant association between abdominal pain and specific inflammation markers. The highest specificity of 92.4% was measured when the abdominal pain persisted more than 6 h after the end of the endoscopic intervention, combined with increased CRP levels and a body temperature of ≥37.5 °C. An elevated CRP value on its own indicates the highest sensitivity of 94.7% for patients with postinterventional abdominal pain exceeding six hours and is an independent PPS predictor. Isolated leukocytosis was not able to indicate or deny PPS properly, with low percentages for both sensitivity and specificity.

In this study, 80 individuals of the whole study population developed a body temperature of at least 37.5 °C, which corresponds to 16.6%. In 45 of these cases, it was not possible to distinguish an isolated increase in temperature with or without an infection, as no other laboratory values were documented. In 27 cases, the elevated body temperature was due to a complication or an acute infectious disease. Ten percent (eight patients) of the patients with a higher body temperature (≥37.5 °C) developed postpolypectomy fever without abdominal pain, so no PPS occurred.

Furthermore, we examined other markers that are not explicitly shown in figures or tables and were not significantly different between the two cohorts: the day of the maximum value of leukocyte count, the day of the maximum value of elevated temperature, the day of the maximum value of CRP, and the presence of PPS in an earlier colonoscopy.

## 4. Discussion

PPS is a heterogeneous postinterventional clinical manifestation after an endoscopic polypectomy that can occur with an increased level of inflammation markers or abdominal pain, fever, or a combination of the mentioned symptoms and signs. Several publications describe this state of abdominal pain, fever, or increased inflammation markers as postpolypectomy syndrome, postpolypectomy coagulation syndrome, postpolypectomy distension syndrome, or simply as transmural burn [2,6,9,12,13]. To date, there is no guideline that clearly defines the symptoms of PPS or the appropriate treatment of it.

The limitations of our study are due to the retrospective study setting, the fact that it was monocentric, the small number in the PPS cohort, and the lack of information regarding the qualities of the pain.

The primary endpoints of this monocentric, retrospective study were the analysis of the clinical course of PPS and potential risk factors and predictors of PPS. Additionally, the defining criteria used to date were examined regarding their importance. The secondary goal was risk stratification for the use of antibiotic prophylaxis in PPS.

The adjusted incidence of PPS in this study was 9.7%. A CRP value ≥ 0.5 mg/dL (94.7%) was found to be the inflammation parameter with the highest sensitivity in predicting PPS and a combination of a CPR value ≥ 0.5 mg/dL and a body temperature ≥ 37.5 °C (92.4%) showed the highest specificity in predicting PPS.

Among our PPS sufferers (30.4%), only every third person had temperatures ≥ 37.5 °C. This is interesting because in most studies reporting a combination of pain and inflammation scores, elevated temperature is an integral part of the definition [14]. Furthermore, the highest temperature value was most frequent (40%) between 38.0 °C and 38.5 °C and reached its maximum (86.7%) after the first postoperative day at the latest.

Leukocytosis (≥10,000 cells/µL) occurred in 54.3% of cases in the range between 10,000 and 15,000 cells/µL in more than every second person. This represents only a moderate increase in the leukocyte count. These values were also reached in most patients (79.3%) by the first postoperative day. While increased temperature occurred more frequently on the day of the operation, this was the case with leukocytosis on the first postoperative day.

CRP values ≥ 0.5 mg/dL were found in 40.0% of PPS sufferers, evenly distributed between low-grade inflammation values defined as a CRP level between 0.5 and 10 mg/dL (20%) and moderate elevated CRP with a range of 10 mg/dL to 20 mg/dL (20%). It usually reached its maximum by the second postoperative day. Of the control cohort, only 20% of the patients developed an increased CRP of more than 0.5 mg/dL. No significant difference was seen between the cohorts.

In the multivariate analysis, localization in the cecum (*p* = 0.021), a serrated morphology (*p* = 0.028), a pedunculated configuration (*p* = 0.003), and the absence of an intraepithelial neoplasia in the grading (*p* = 0.009) could be identified as significant independent risk factors for PPS.

An increased risk of PPS in the right side of the colon, as demonstrated in this study for the cecum, seems plausible given that the right side of the colon has a thinner intestinal wall than the left side [15,16]. The thermal energy applied via electrocautery can thus more easily and quickly cause damage to the muscularis propria, possibly the serosa, or rather lead to a local inflammatory reaction [17,18].

Thermal wall damage and increased release of inflammatory transmitters in the resection area are convincing theories for the development of PPS. There was insufficient evidence for a bacteria-associated inflammatory reaction in this trial, as none of the patients in the PPS cohort had a positive blood culture. We consider that the cause of PPS is an inflammatory burden with a potential invasion of the gut microbiome through a non-intact colon wall. Our data correlate with other published data from Min et al. [19] and Lee et al. [18] with a bacteremic incidence of 3.6% and 5.3%. In those studies, the low bacteremic incidence was discussed as probably contaminated blood cultures.

A risk stratification for the use of prophylactic antibiotic treatment in PPS could not be evaluated in our work because the sample size was too small, hence usable results could not be expected. Muro et al. as well as Shi et al. could not demonstrate the significance of prophylactic treatment with antibiotics to prevent PPS after polypectomy [20,21].

Serrated adenomas represent 5.8–11% of all colon and rectal adenomas [22,23,24,25]. One possible explanation of the pathophysiology of serrated adenomas in the PPS involves the difficulty in delineating the margins of the serrated lesions from the normal mucosa [26]. According to the NICE classification, serrated sessile lesions and hyperplastic polyps show a different surface structure than the adjacent regular colonic mucosa, but these differences in surface color, structure, and vascular status are significantly smaller than in the “classic” adenoma [27]. The identification of the edge is therefore more difficult in serrated lesions than in known adenomas. Furthermore, unclear marginal conditions can lead to incomplete resections, thereafter requiring an immediate full resection, whereby the mucosa is more severely damaged compared to a single resection.

It was not expected that the absence of an IEN in the resected polyps would present a significant difference between the two cohorts and lead to a higher rate of PPS. We did not find other publications that demonstrated this risk factor.

A pedunculated configuration (corresponding to PC 0-Ip) of the polyps is our fourth significant risk factor for the presence of PPS. A possible explanation could be that during EMR, the polyps are usually injected, whereupon the polyps lift up into the lumen. Sessile or non-polypoid polyps are flatter, whereas pedunculated polyps have a larger longitudinal axis. This can lead to the head of the polyp touching the opposite intestinal mucosa, transmitting the applied electrical energy and leading to wall damage there [28]. Waye et al. described that this local wall injury was not sufficient for a perforation but attested that a slight whitish discoloration on the opposite side as a sign of the transferred damage was not uncommon [29].

Risk factors from the comparative literature such as female sex [5,17,30,31], arterial hypertension [32], submucosal fibrosis [5], a non-polypoid polyp configuration [32], as well as polyp size [8,17,32] and duration of surgery [30], did not show any significant differences in our analysis.

Individual papers indicate that PPS occurs less frequently under cold snare resection than after the classic hot snare resection method. While Fatima et al. did not have a single case of PPS documented after cold snare polyp resection [33], Guo et al. demonstrated a significantly higher incidence of PPS with hot snare EMR than with cold snare EMR in their propensity score-matching observational study [34]. These observations are supported by the results of a prospective study by Suzuki et al. They showed a deeper wound from cold snare vs. hot snare polypectomy—initially developed directly after the removal of the polyp, but the situation reversed from the first postoperative day and the wounds caused by cold snare were significantly more superficial afterward [35].

Further prospective studies should investigate the effect of cold snare resection on PPS, since according to current studies, the cold snare technique could lead to a reduction in PPS while at the same time confirming the theory of thermal wall damage.

We developed a diagnostic and therapeutic algorithm for cases of suspected PPS, which is shown in Figure 8. Most of the algorithm pathways are standard of care, like operation in the case of detected free intraabdominal air or antibiotic treatment if a covered perforation is suspected. Figure 7 does not suggest a specific therapeutic procedure for suspected PPS, as there is a lack of evidence for it [36,37]. Prospective multicentric trials are needed to validate and confirm the relevance and evidence of this algorithm.

To summarize, it is important to be aware of PPS and to know that PPS is often a self-limiting disease, which usually can be treated symptomatically. Hence, it can be classified as a minor AE after polypectomy, compared to an acute perforation with peritonism and the need for an operation.

## 5. Conclusions

Four significant independent risk factors could be elucidated, which were associated with a higher risk of developing PPS. These are serrated polyp morphology, polyp localization in the cecum, polypoid configurated adenomas, and the absence of intraepithelial neoplasia in the resected polyps. Furthermore, the combination of CRP and a body temperature of ≥37.5 °C is a predictor, as it showed a specificity of 92.4% and CRP alone showed a sensitivity of 94.7% in detecting PPS in patients with persisting abdominal pain in the anatomical area of polyp resection for at least six hours. To summarize, we recommend a diagnostic triad consisting of persistent local abdominal pain for more than six hours, as well as both an increased CRP level and an elevated body temperature of ≥37.5 °C within 24 h since the polypectomy procedure to define PPS.

Further prospective trials are needed to identify other predictors and risk factors of PPS and the efficacy of preventing PPS via the administration of antibiotics, antiphlogistic medication, or intravenous fluids.

## Figures and Tables

**Figure 1 diagnostics-14-00127-f001:**
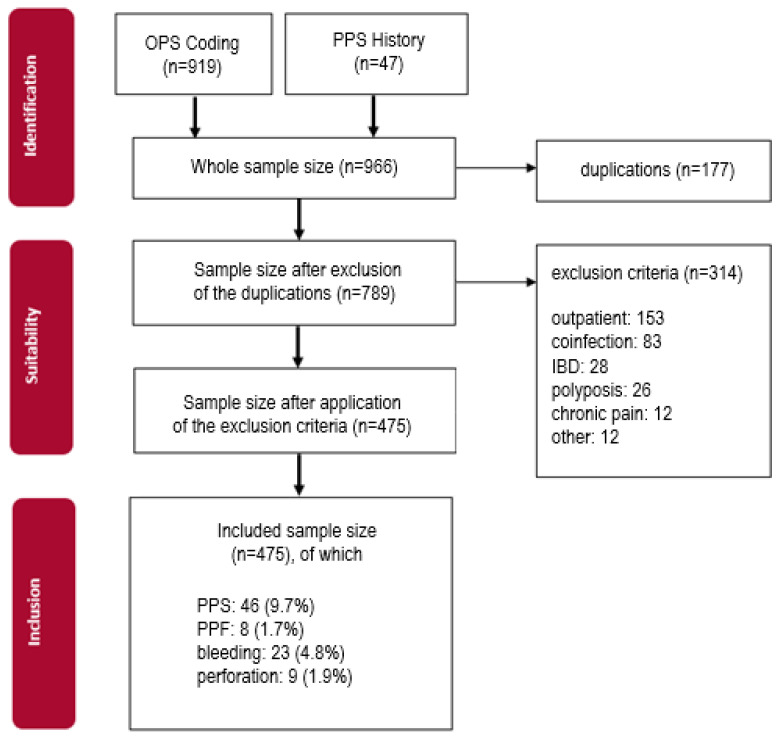
Flowchart of patient identification and inclusion.

**Figure 2 diagnostics-14-00127-f002:**
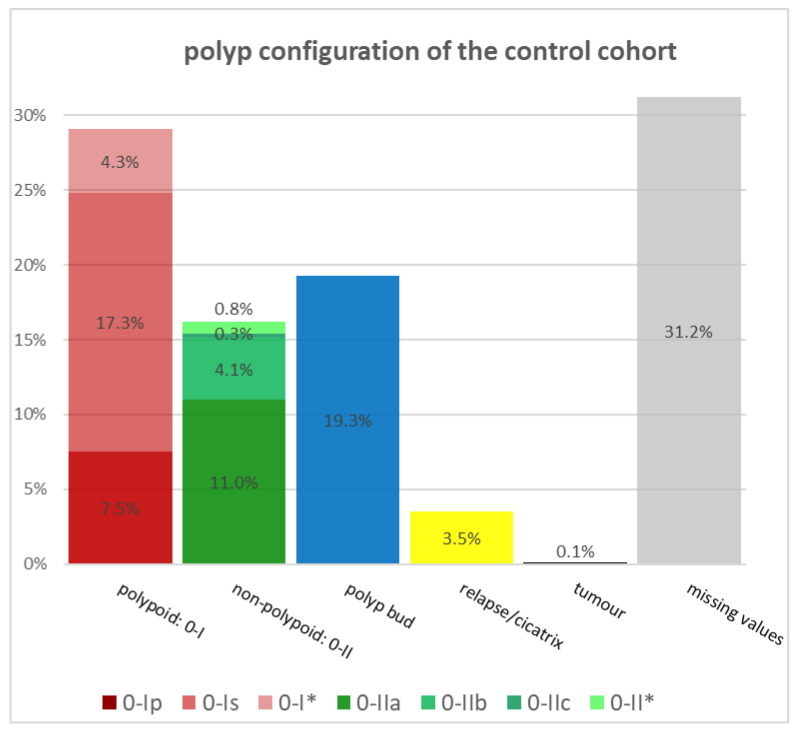
Polyp configuration of the control cohort (Paris classification).

**Figure 3 diagnostics-14-00127-f003:**
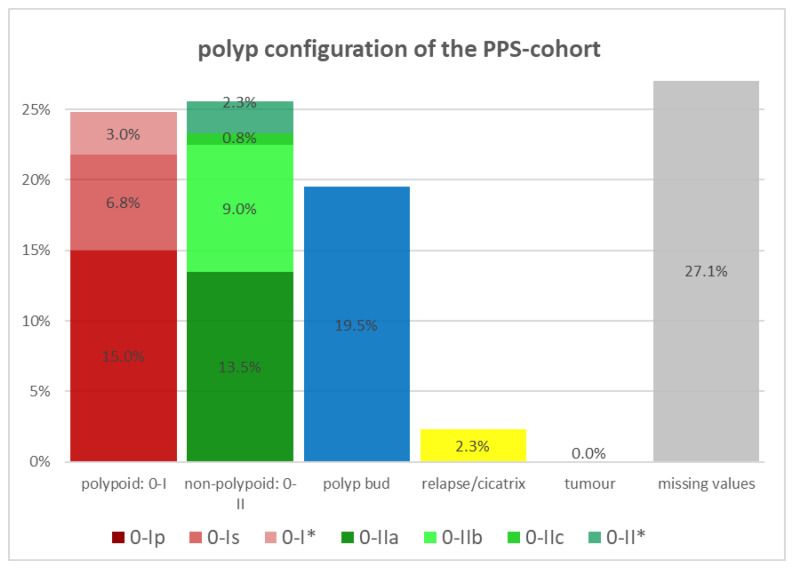
Polyp configuration of the PPS cohort (Paris classification).

**Figure 4 diagnostics-14-00127-f004:**
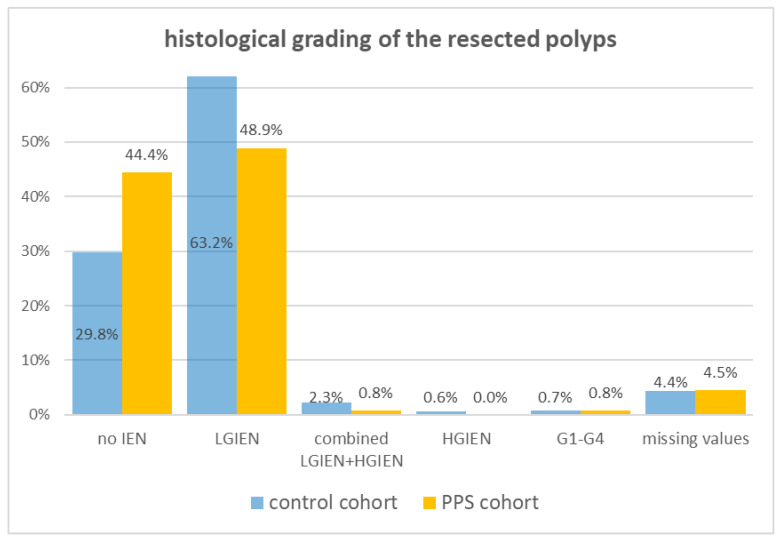
Histological grading of the resected polyps.

**Figure 5 diagnostics-14-00127-f005:**
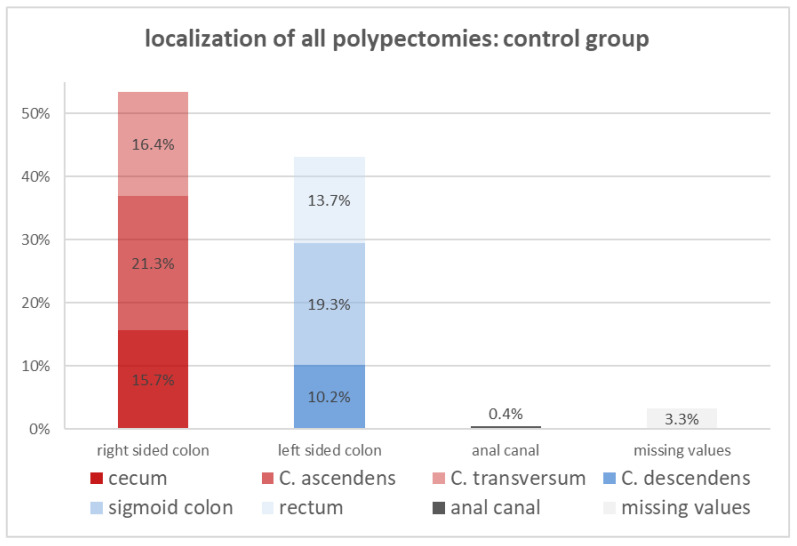
Overview of localization of polypectomies in the control group (*n* = 1156).

**Figure 6 diagnostics-14-00127-f006:**
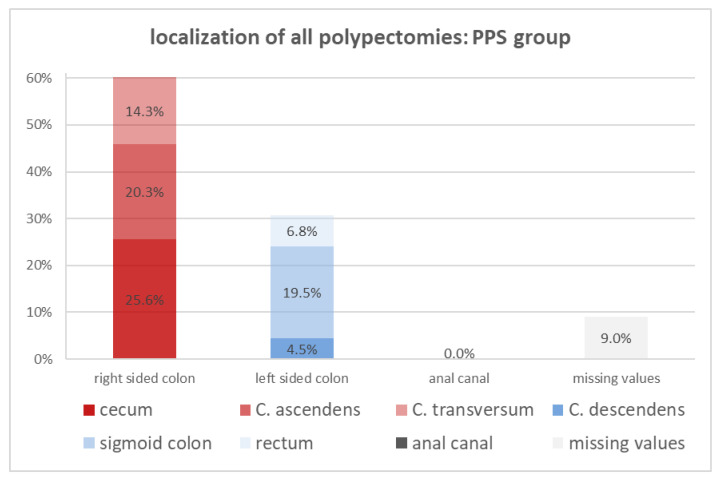
Overview of localization of polypectomies in the PPS group (*n* = 133).

**Figure 7 diagnostics-14-00127-f007:**
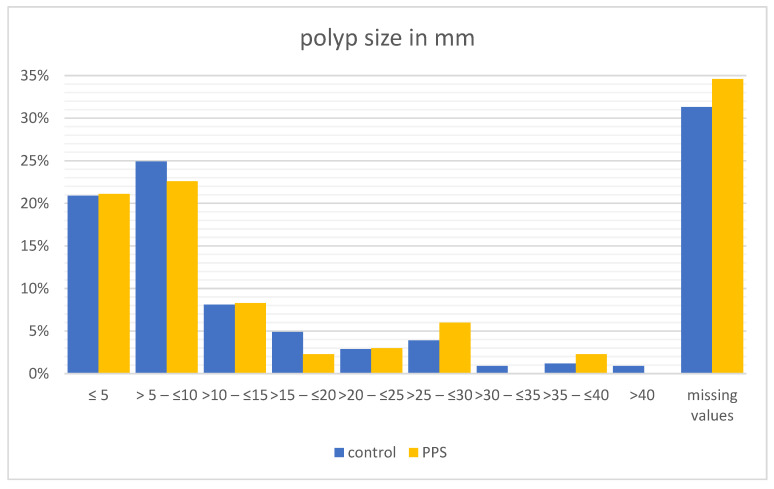
Polyp size in mm in control group and PPS group.

**Figure 8 diagnostics-14-00127-f008:**
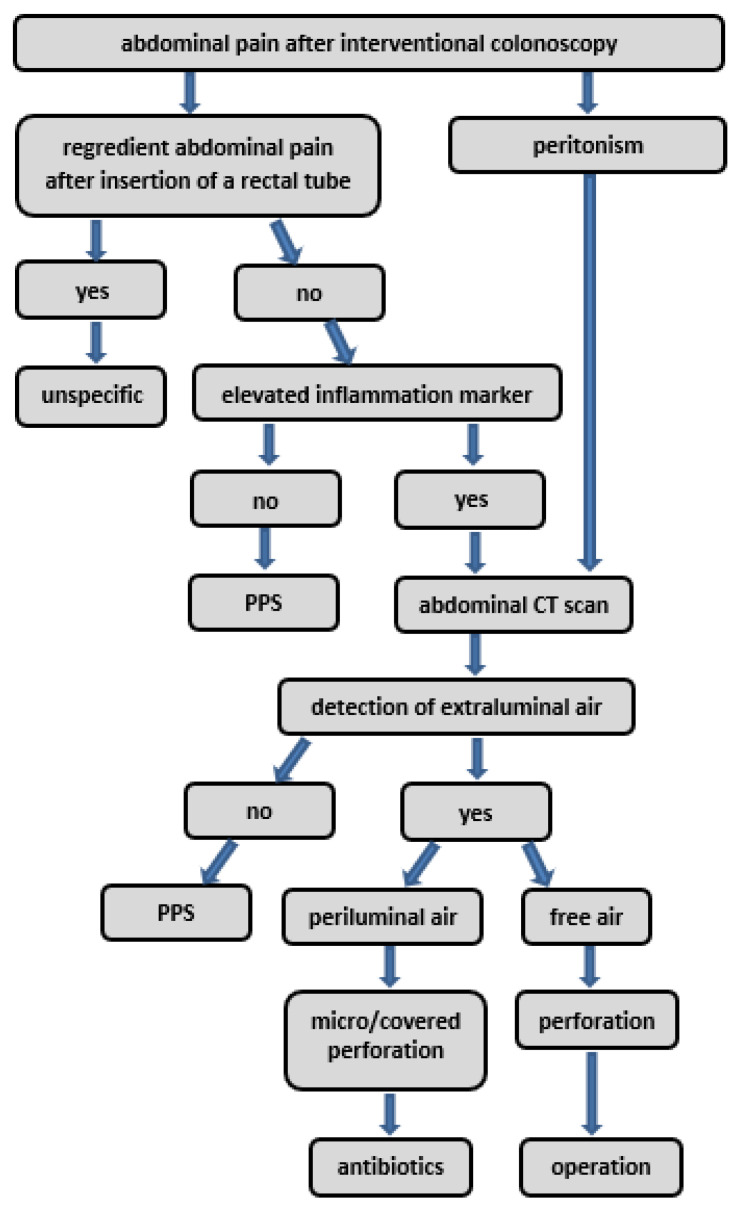
Diagnostic and therapeutic algorithm for suspected PPS.

**Table 1 diagnostics-14-00127-t001:** Basic and clinical characteristics of the PPS and the control group.

	Control	PPS	*p* Values
**Cohort Size, *n***	***n*_1_ = 429**	***n*_2_ = 46**	
** *patient characteristics* **			
**sex, *n* (%)**			
female	154 (35.9)	20 (43.5)	0.28
male	275 (64.1)	26 (56.5)	0.42
**age, mean ± SD**	68.8 ± 11.1	65.7 ± 11.1	0.68
**BMI**			
*n*_1_: median (IQR)	26.1 (24.0–29.4)		
*n*_2_: mean ± SD		27.6 ± 4.9	
**pre-existing diseases, *n* (%)**	279 (65.0)	28 (60.9)	0.36
coronary heart disease	63 (14.7)	5 (10.9)	0.29
diabetes mellitus	76 (17.7)	7 (15.2)	0.71
arterial hypertension	205 (47.8)	25 (54.3)	0.32
kidney insufficiency	46 (10.7)	2 (4.3)	0.18
liver insufficiency	29 (6.8)	4 (8.7)	0.45
COPD	19 (4.4)	3 (6.5)	0.51
autoimmune disorder	17 (4.0)	1 (2.2)	0.34
immunosuppressive disease	9 (2.1)	2 (4.3)	0.29
multimorbidity	126 (29.4)	13 (28.3)	0.75
**long-term analgesic medication, *n* (%)**	56 (13.1)	7 (15.2)	0.21
**nicotine abuse, *n* (%)**	136 (31.7)	8 (17.4)	0.09
yes	89 (65.4)	6 (75.0)	0.23
no	47 (34.6)	2 (25.0)	0.17
** *peri-procedural aspects* **			
**antibiotic pretreatment, *n* (%)**	15 (3.5)	1 (2.2)	0.34
**antibiotic treatment, *n* (%)**	35 (8.2)	11 (23.9)	0.18
**blood cultures, *n* (%)**	8 (1.9)	4 (8.7)	0.22
negative	6 (1.4)	4 (8.7)	0.30
positive	2 (0.5)	0	0.56
**periluminal air, *n* (%)**			
negative	19 (4.4)	6 (13.0)	0.16
positive	0	1 (2.2)	0.43
**adverse events, *n* (%)**			
perforation	9 (2.1)		
periinterventional bleeding	59 (13.8)		
postinterventional bleeding	23 (5.4)		
PPF	8 (1.8)		
other	2 (0.5)		

BMI: body mass index. COPD: chronic obstructive pulmonary disease. PPF: Postpolypectomy fever.

**Table 2 diagnostics-14-00127-t002:** (**a**) Endoscopic characteristics of the patients. (**b**) Endoscopic characteristics of the polyp samples.

(a)			
	**Control**	**PPS**	***p* Value**
**sample size, *n***	**429**	**46**	
**duration of intervention (min)**			
median (IQR)	45 (31.3–61.5)	52 (40–70)	0.24
**antibiotic during inpatient care, *n* (%)**	35 (8.2)	11 (23.9)	0.12
**particularities, *n* (%)**	52 (12.1)	6 (13)	0.67
**APC, *n* (%)**	84 (19.6)	8 (17.4)	0.54
**clip application, *n* (%)**	193 (45.0)	20 (43.5)	0.72
**relevant diverticulosis, *n* (%)**	143 (33.3)	18 (39.1)	0.34
**(b)**			
**number of polyps, *n***	**1156**	**133**	
median (IQR)	2.84 (1–4)	2.98 (1–4)	0.53
**localization, *n* (%)**	**1122 (97.0)**	**121 (91)**	
cecum	182 (15.7)	34 (25.6)	0.45
colon ascendens	246 (21.3)	27 (20.3)	0.51
colon transversum	190 (16.4)	19 (14.3)	0.34
colon descendens	118 (10.2)	6 (4.5)	0.29
sigmoid colon	223 (19.3)	26 (19.5)	0.75
rectum	158 (13.7)	9 (6.8)	0.21
anal canal	5 (0.4)	0	0.69
**result of resection, *n* (%)**	**1067 (92.3)**	**121 (91)**	
failure	1 (0.1)	0 (0)	0.76
en bloc	827 (71.5)	94 (70.7)	0.82
piecemeal	202 (17.5)	25 (18.8)	0.53
incomplete	30 (2.6)	2 (1.5)	0.47
**method of intervention, *n* (%)**	**1052 (91)**	**121 (91)**	
forceps	381 (33.0)	37 (27.8)	0.26
EMR	716 (61.9)	88 (66.2)	0.24
EMR combined	46 (6.4% of EMR)	8 (10)	
ESD	0	0	1.00
EFTR	15 (1.3)	1 (2.2)	0.64
EFTR combined	9 (60% of EFTR)	0	

APC: Argon Plasma Coagulation. EMR: endoscopic mucosa resection. ESD: endoscopic submucosa dissection. EFTR: endoscopic fully thickness resection.

**Table 3 diagnostics-14-00127-t003:** Univariate statistics test for possible risk factors using the Chi-square test.

	Odds Ratio (95% CI)	*p*-Value ≤ 0.05
**female sex, *n* (%)**	1.374 (0.742–2.542)	0.310
**pre-existing diseases, *n* (%)**		
- arterial hypertension	1.301 (0.707–2.395)	0.397
**duration of the procedure**		
- ≥50 min	1.723 (0.925–3.208)	0.083
**relevant diverticulosis**	1.277 (0.683–2.386)	0.443
**localization, *n* (%)**		
- right side of colon	1.653 (0.774–3.536)	0.190
- cecum	2.021 (1.096–3.727)	**0.022**
- colon ascendens	1.207 (0.653–2.231)	0.548
**piecemeal resection, *n* (%)**	1.356 (0.737–2.496)	0.326
**polyp configuration, *n* (%)**		
- pedunculated	2.470 (1.229–4.966)	**0.009**
- not polypoid	1.527 (0.809–2.881)	0.189
- flat	2.233 (0.870–5.731)	0.087
**polyp morphology, *n* (%)**		
- not polypoid	1.568 (0.852–2.2885)	0.146
- serrated	3.716 (1.556–8.875)	**0.002**
**grading, *n* (%)**		
- absent IEN	2.700 (1.416–5.148)	**0.002**
**submucosal fibrosis, *n* (%)**	0.430 (0.101–1.840)	0.242

**Table 4 diagnostics-14-00127-t004:** Multivariate linear regression defining four independent risk factors for PPS.

Coefficients ^a^							
	Unstandardized Coefficients		Standardized Coefficients			95% Confidence Interval for B	
Model	B	Std. Error	Beta	t	Sig.	Lower Bound	Upper Bound
(Constant)	0.016	0.021		0.759	0.448	−0.026	0.058
no EIN	0.072	0.028	0.121	2.615	0.009	0.018	0.126
pedunculated	0.112	0.037	0.136	3.025	0.003	0.039	0.185
serrated	0.123	0.056	0.102	2.206	0.028	0.013	0.232
cecum	0.065	0.028	0.105	2.324	0.021	0.010	0.120

^a^ Dependent Variable: PPS/Control.

**Table 5 diagnostics-14-00127-t005:** Potential PPS predictors.

Sample Size, *n*	*n*_1_ = 429	*n*_2_ = 46
body temperature (°C),		
*n*_1_ = median + IQR (min; max)	36.9 (36.5–37.3)	
*n*_2_ = mean ± SD		37.2 ± 0.8
leukocytes (cells/µL), median (IQR)	9200 (7115–11,288)	10,885 (8163–13,708)
thrombocytes, (10^3^/µL), mean ± SD	210,415 ± 70,290	199,262 + 77,848
CRP (mg/dL), median (IQR)	1.9 (1; 4.3)	9.4 (3.2–14.8)

CRP: C reactive protein.

**Table 6 diagnostics-14-00127-t006:** Sensitivity and specificity of inflammation markers.

	Abdominal Pain ≤ 6 h		Abdominal Pain ≥ 6 h	
	Sensitivity	Specificity	Sensitivity	Specificity
**CRP** (≥0.5 mg/dL)	**83.3%**	33.3%	**94.7%**	35.3%
**Leukocytosis**	51.1%	62.3%	59.5%	63%
**≥37.5 °C**	26.9%	84.2%	31.1%	84.5%
**CRP + ≥37.5 °C**	13.6%	**91.5%**	22.5%	**92.4%**

CRP: C reactive protein.

## Data Availability

Due to ethical restrictions we can not share the data.

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
