# Peer review of "Analysis of Predictors and Risk Factors of Postpolypectomy Syndrome"

_diagnostics, 2024, doi:10.3390/diagnostics14020127_

Round 1

Reviewer 1 Report

Comments and Suggestions for Authors

In this single center retrospective study about post-polypectomy syndrome (PPS), Fusco et al showed that such event was linked to serrated polyps, adenoma, caecal site and intraepithelial neoplasia. Main comments:

1) Page 2 lines 74-76: the definition of PPS should be supported by a reference.

2) Figure 1: arrows to “duplications” and “exclusion criteria” refer to the wrong boxes.

3) Tables 1-2: please add p values.

4) How many patients were hospitalized for polypectomy?

5) Please check grammar (e.g. positive/negativ).

6) What do authors mean for “particularities”? (Table 2)

7) What do authors mean for “polyp bud”? This is a definition not described in Paris classification.

8) Polyp size was described in lines 192-197, but it was not considered in univariate and multivariate. Therefore such analysis should be re-done by considering such variable.

9) Abdominal pain could be due to over-insufflation of the colon rather than PPS. Please discuss.

10) The finding described in line 301 (polypoid shape as a risk factor) is in contradiction with literature data (ref. 29), therefore a more convincing discussion is necessary.

Comments on the Quality of English Language

See above

Reviewer 2 Report

Comments and Suggestions for Authors

This is a retrospective study on postpolycpectomy syndrome, in which potential risks of presenting PPS are analyzed.

In general, I consider that the study is well structured, presents relevant and interesting information, and the discussion is in accordance with the results obtained, as well as the conclusions. I only have a few points that I think are important to adjust in the article:

The bibliographic reference corresponding to the statements of the paragraph that begins on line 45 must be placed.

Change the term person to patient on line 90.

In Table 2, I consider that putting asterisks to differentiate the data obtained from the biopsy is somewhat confusing and it would be worth separating them, either by titling a subsection of the table or in a separate table.

I consider that there would still be a lack of evidence to propose an algorithm like the one suggested by the authors, but in case they consider proposing it, it is necessary to justify it not only with the data of this article but also with bibliographic data that complement it, as well as continue leaving the clarification that it must be validated. If you do not comply with this, I consider it better to withdraw it.

Round 2

Reviewer 1 Report

Comments and Suggestions for Authors

Regarding point 3, it is not enough to write "p>0.05". Authors must add the exact p value. They may add a column for this.

Polyp bud is an awkward term. I suggest to replace with "diminutive polyp".

Point 8: Authors can add the figure by replacing figure 7, which is less important. Figure 7 may be added as supplementary file.

All other answers were OK.

After such changes, I believe that the paper may be accepted.

Round 3

Reviewer 1 Report

Comments and Suggestions for Authors

Answers were fine